# Earth System Model Evaluation Tool (ESMValTool) v2.0 – diagnostics for extreme events, regional and impact evaluation and analysis of Earth system models in CMIP

Katja Weigel<sup>1,2</sup>, Lisa Bock<sup>2</sup>, Bettina K. Gier<sup>1,2</sup>, Axel Lauer<sup>2</sup>, Mattia Righi<sup>2</sup>, Manuel Schlund<sup>2</sup>, Kemisola Adeniyi<sup>1,2</sup>, Bouwe

5 Andela<sup>3</sup>, Enrico Arnone<sup>4,5</sup>, Peter Berg<sup>6</sup>, Louis-Philippe Caron<sup>7</sup>, Irene Cionni<sup>8</sup>, Susanna Corti<sup>4</sup>, Niels Drost<sup>3</sup>, Alasdair Hunter<sup>7</sup>, Llorenç Lledó<sup>7</sup>, Christian Wilhelm Mohr<sup>9,10</sup>, Aytaç Paçal<sup>2</sup>, Núria Pérez-Zanón<sup>7</sup>, Valeriu Predoi<sup>11</sup>, Marit Sandstad<sup>9</sup>, Jana Sillmann<sup>9</sup>, Andreas Sterl<sup>12</sup>, Javier Vegas-Regidor<sup>7</sup>, Jost von Hardenberg<sup>13,4</sup>, and Veronika Eyring<sup>2,1</sup>

<sup>1</sup>University of Bremen, Institute of Environmental Physics (IUP), Bremen, Germany.
 <sup>2</sup>Deutsches Zentrum für Luft- und Raumfahrt (DLR), Institut für Physik der Atmosphäre, Oberpfaffenhofen, Germany
 <sup>3</sup>Netherlands eScience Center (NLeSC), Amsterdam, the Netherlands
 <sup>4</sup>Institute of Atmospheric Sciences and Climate, Consiglio Nazionale delle Ricerche (ISAC-CNR), Italy
 <sup>5</sup>Department of Physics, University of Torino, Italy
 <sup>6</sup>Hydrology research unit, Swedish Meteorological and Hydrological Institute (SMHI), Sweden
 <sup>7</sup>Barcelona Supercomputing Center (BSC), Barcelona, Spain
 <sup>8</sup>Agenzia nazionale per le nuove tecnologie, l'energia e lo sviluppo economico sostenibile (ENEA), Rome, Italy
 <sup>9</sup>CICERO - Center for International Climate Research, Oslo, Norway

- <sup>10</sup>Now at: Division for Forestry and Forest Resources, The Norwegian Institute of Bioeconomy Research (NIBIO), Ås, Norway
   <sup>11</sup>NCAS Computational Modelling Services (CMS), University of Reading, Reading, UK
   <sup>12</sup> Roval Netherlands Meteorological Institute (KNMI), de Bilt, The Netherlands
- <sup>13</sup>Department of Environment, Land and Infrastructure Engineering, Politecnico di Torino, Turin, Italy

Correspondence to: Katja Weigel (weigel@iup.physik.uni-bremen.de)

**Abstract.** This paper complements a series of now four publications that document the release of the Earth System Model Evaluation Tool (ESMValTool) v2.0. It describes new diagnostics on the hydrological cycle, extreme events, impact assessment, regional evaluations, and ensemble member selection. The diagnostics are developed by a large community of

- scientists aiming to facilitate the evaluation and comparison of Earth System Models (ESMs) which are participating in the Coupled Model Intercomparison Project (CMIP). The second release of this tool aims to support the evaluation of ESMs participating in CMIP Phase 6 (CMIP6). Furthermore, data sets from other models and observations can be analysed. The diagnostics for the hydrological cycle include several precipitation and drought indices, as well as hydroclimatic intensity and indices from the Expert Team on Climate Change Detection and Indices (ETCCDI). The latter are also used for identification
- of extreme events and for impact assessment, and to project and characterize the risks and impacts of climate change for natural and socio-economic systems. Further impact assessment diagnostics are included to compute daily temperature ranges and capacity factors for wind and solar energy generation. Regional scales can be analysed with new diagnostics implemented for selected regions and stochastic downscaling. ESMValTool v2.0 also includes diagnostics to analyse large multi-model

ensembles including grouping and selecting ensemble members by user specified criteria. Here, we present examples for their

capabilities based on the well-established CMIP Phase 5 (CMIP5) data set.

# **1** Introduction

Climate change is affecting the Earth system in many different ways. To be able to assess the impacts of climate change on society and to develop strategies for mitigation and adaptation, a detailed knowledge of the climate system and the key processes driving climate change is necessary. This is particularly the case for changes in the hydrological cycle and climate extreme events, both having direct consequences on ecosystems and society (Eyring et al., 2020). With rising greenhouse gas concentrations the hydroclimatic regime is expected to change (Giorgi et al., 2019). As the intensity and distribution of precipitation determines the availability of fresh water in a certain region, it is also related to the severity of hazardous events such as flooding or droughts. The impact of extreme events on many socio-economic factors increases with their severity, but the rare occurrence of these events makes an assessment of the effect of climate change on such events challenging (Zhang et

al., 2011). Especially compound events, caused by a combination of processes on multiple spatial and temporal scales, lead to severe impacts (Zscheischler et al., 2018).

Changes in climate can alter both the strength and the probability of extreme events (Seneviratne et al., 2012; IPCC, 2012). For various extreme events an increase in severity and frequency was observed in the past decades and is expected with rising temperatures, as, for instance, for warm temperature extremes (Alexander, 2016). With rising temperatures an increase is also

- expected for the amount of precipitation. For wet precipitation extremes this increase is expected to happen faster than for the total wet-day (days with precipitation > 1 mm) precipitation (Sillmann et al., 2013b). Several studies project that dry regions become drier and wet regions wetter (Martin, 2018; Greve et al., 2014), which is expected to result in an increase in both wet and dry extreme events, depending on the region. This tendency was highlighted by a general increase of the hydroclimatic intensity, which gives a joint measure of dry and wet conditions in a warming climate (Giorgi et al., 2011). Studies by Donat
- et al. (2019) and Pfahl et al. (2017) show an increase in observed precipitation extremes in humid regions whereas there is no clear indication on the change of precipitation extreme events in arid regions. The impact of different climate forcers such as greenhouse gases and aerosols on droughts remains to be understood in more detail (Marvel et al., 2019).

Although the climate system is of global extent, its manifestations have regional and local impacts (IPCC, 2014a). Especially for regional climate changes, robust projections require not only an understanding of the underlying physics and internal

variability but also a reduction of model biases is essential (Xie et al., 2015). If model biases are corrected without considering the underlying physical processes, however, downscaling of ESM results to regional scales can result in unwanted artefacts (Maraun et al., 2017). Observed changes on the regional scale depend to a large extent on atmospheric dynamics, therefore the signal of climate change is often smaller than the internal variability (Deser et al., 2012) while large differences are found in the modelled future scenarios (Shepherd, 2014). Stochastic downscaling of precipitation can aid in this direction as the fields

- at regional scale are derived from the spectral properties of the fields at large scale, with an ability in the reproduction of extremes even over complex orography (Rebora et al. 2006; D'Onofrio et al. 2014; Terzago et al., 2018). Model ensembles can be used to quantify uncertainties in climate change projections due to internal variability (Xie et al., 2015) and clustering analysis to inter-compare and group ensemble members based on similar characteristics and select the most representative ones, going beyond the biases of individual models (Straus et al., 2007).
- The Earth System Model Evaluation Tool (ESMValTool) version 2.0 (v2.0) includes diagnostics and performance metrics for the analysis and evaluation of ESMs with observations. It is developed by a large community, which involves more than 150 scientists from over 60 institutions. Figures and other output produced by the tool include full provenance information to allow for traceability and reproducibility of the results. The main focus is on the analysis of ESM simulations from the Coupled Model Intercomparison Project (CMIP) of the World Climate Research Programme (WCRP). CMIP started in 1995 (Meehl et
- al., 2000) with the aim of providing scientists with comparable coupled model runs based on standardized boundary conditions (Covey et al., 2003). CMIP results from Phase 5 (CMIP5) (Taylor et al., 2012) are the basis for many assessments in the IPCC's Fifth Assessment Report (AR5) (IPCC 2013). Now, data from Phase 6 (CMIP6) (Eyring et al., 2016) are available. With every phase of CMIP the volume of data increases: for CMIP6 a total data volume of about 20 to 40 PB is expected. This emphasizes the need for a fast and comprehensive tool like the ESMValTool (v2.0) to evaluate these model results. In this
- work, the diagnostics which focus on climate impacts are described and their output using the well-established CMIP5 data is shown.

In this study we present diagnostics included in the ESMValTool specifically for the analysis of the hydrological cycle, extreme events, climate impacts, multi-model ensemble member sub-selection, and regional model evaluation. This article completes a series of publications documenting ESMValTool v2.0: Righi et al. (2020) describes the technical aspects, (Eyring et al.,

2020) the new large-scale diagnostics, and (Lauer et al., 2020) emergent constraints and diagnostics for future projections from ESMs in CMIP.

This paper is organized as follows: Section 2 describes the model and observation data used. Section 3 presents the ESMValTool recipes for the analyses of hydroclimatic intensity, droughts, extreme events, model impact evaluation, multimodel ensemble member sub-selection, and regional model evaluation. It also describes use of the ESMValTool as a post-

90 processing tool for further downscaling applications. Section 4 closes with a summary.

# 2 Models and observations

ESMValTool v2.0 was developed particularly for the analysis of CMIP data (Righi et al., 2020). This work mainly presents results based on the well-established CMIP5 model ensemble, but other model output and observational data, e.g. provided by observations for Model Intercomparison Project (obs4MIPs; Teixeira et al., 2014; Waliser et al., 2020), can also be analysed.

As in version v1.0 (Eyring et al., 2016) ESMValTool v2.0 expects input data to be in a Climate and Forecast (CF) Metadata

compliant Network Common Data Format (netCDF) following the Climate Model Output Rewrite (CMOR) standard. The detailed requirements for CMOR can be found in these tables (<u>http://pcmdi.github.io/cmor-site/tables.html</u>). For the recipes described here, European Centre for Medium-Range Weather Forecasts (ECMWF) ERA-Interim and Climatic Research Unit (CRU) reanalysis data are used for the evaluation of the model results. Table 1 lists these data in case they are used for a recipe.

- These datasets should be seen as examples as they can easily be replaced by other reanalysis or observational datasets. Reformatting scripts with downloading instructions are provided with the ESMValTool v2.0 to convert many observational datasets to the CMOR standard. A list of observational datasets available can be found in Righi et al. (2020) and in the user's guide at https://docs.esmvaltool.org/en/latest/input.html#supported-datasets", where it is updated for newly included datasets. For ECMWF ERA5 a "cmorization-on-the-fly" is implemented, which works on the ERA5 netCDF data directly and does not
- require prior reformatting.

# 3 Overview of recipes included in ESMValTool v2.0

This section describes the new and extended ESMValTool v2.0 recipes for analysis of extreme events, regional model output, and for applying ESM output in the assessments of the impact of climate change as well as to carry out model ensemble subselection. In ESMValTool v2.0, a recipe is a \*.yml file used to define the diagnostics and performance metrics to apply to the

110 simulation output, as well as the datasets and variables used. The ESMValTool is started from the command line, using for example:

# esmvaltool run esmvaltool/recipes/examples/recipe\_python.yml

Where "esmvaltool/recipes/examples/recipe\_python.yml" is one possible recipe. Instead of this example, any other recipe provided with the ESMValTool or created by the user can be used. For a more detailed instructions how to run the tool and modify or create recipes, see the documentation at https://docs.esmvaltool.org/.

- modify or create recipes, see the documentation at https://docs.esmvaltool.org/. In the following, the recipes are briefly described and illustrated with example figures using CMIP5 data. All recipes presented in this work are summarized in Table 1, which includes a short description, together with the analysed variables used, the applied diagnostics and their purpose, as well as the references the diagnostics are based on. Because the online documentation for the ESMValTool v2.0 at <u>https://docs.esmvaltool.org/</u> was written simultaneously to this paper by the same authors, there
- is a considerable overlap to this not peer-reviewed document.

Section 3.1 describes recipes for the hydrological cycle, including indices for hydroclimatic intensity and drought detection. In section 3.2 recipes for other extreme events are presented. Recipes for model impact assessment are described in Section 3.3, and recipes for regional model evaluation in Section 3.4. Section 3.5 presents a recipe for the sub-selection of multi-model ensemble members.

# 125 3.1 Hydrological cycle

## 3.1.1 Hydroclimatic intensity and related indices

The Earth's hydrological cycle is a key element of the climate system with important impacts on the society. For example, the intensity and distribution of precipitation determines the abundance or scarcity of fresh water in a certain region. It is also related to the severity of hazardous events such as flooding or droughts. Several studies have shown an acceleration of the

- hydrological cycle and an intensification of both dry and wet extremes in a warming climate (IPCC, 2013). A simple investigation of total precipitation-related quantities can hide some of the most relevant aspects of the hydrological cycle and its extremes, which can be highlighted through the joint use of the concept of hydroclimatic intensity and related indices (e.g., Giorgi et al., 2014). The hydroclimatic intensity (Giorgi et al., 2011), derived as the product of mean daily precipitation and dry spell length normalized over a reference period, offers a joint view of both dry and wet conditions, allowing to uniquely
- quantify the response in the intensity of the hydrological cycle in a changing climate. The *hyint (hydroclimatic intensity)* diagnostic was developed to calculate several indices for hydroclimatic and climate extremes and allow a multi-index evaluation of climate models.

The *recipe\_hyint.yml* calculates six indices for evaluating the to global warming response of the hydrological cycle including both, wet and dry extremes. The indices are selected according to Giorgi et al. (2014), including the simple precipitation

- intensity index (SDII), the maximum dry spell length (DSL) and wet spell length (WSL), the hydroclimatic intensity index (HY-INT, calculated as normalized DSL times normalized SDII), which is a measure of the intensity of the hydroclimatic cycle compared to a reference period (Giorgi et al., 2011), and the precipitation area (PA), i.e. the area over which precipitation occurs at any given day (Giorgi et al., 2014). The *recipe\_hyint\_extreme\_events.yml* can also ingest the 27 temperature and precipitation-based Expert Team on Climate Change Detection and Indices (ETCCDI) (Zhang et al., 2011) calculated by the
- recipe\_extreme\_events.yml to produce a multi-index analysis (see Sec. 3.2 for further details). The diagnostics perform a subsequent analysis calculating time series and trends of the selected indices for predefined continental areas, normalized to a reference period. The linear model (lm) function of R is used to calculate trends. Statistical significance is tested based on a Student's t-test under a non-null coefficients hypothesis. Trend coefficients and their statistics including standard error, p-value and precipitation above the 95% percentile of the reference distribution are stored. The recipe created several plots,
- amongst others global and regional maps, timeseries with spread, trend lines and summary plots of trend coefficients. Results are stored in netCDF files including relevant information such as normalization functions and thresholds, and as figures. Figures 1 and 2 show examples of an analysis performed with the hyint diagnostic. A map of the HY-INT index (Fig. 1) calculated from EC-EARTH model data shows the projected average HY-INT compared to the reference period (1976-2005): hydroclimatic intensity is projected to greatly increase in some regions (e.g., eastern South America, northern Africa and the
- Arabian peninsula) and to decrease over other regions (e.g., Antarctica, Greenland, central and North-East Asia, central Africa and western and northern South America) with large areas showing only moderate changes. Trends shown in Fig. 2 exhibit a

relatively low inter-model spread for HY-INT. The projected increase in HY-INT seen for all models with values ranging around 10% per century models (also reflected as large geographical patterns) can also be seen in the precipitation intensity (SDI) and heavy precipitation indices (R95), the latter with an increase spread between 10 and 30% per century. Precipitation

area (PA) is projected to increase by most models, whereas for projected changes in the dry spell length (DSL) and especially in the wet spell length (WSL) models do not agree on the sign of the projected changes, reflected also in high geographical variability (not shown).

# **3.1.2 Droughts**

Three main types of droughts can be separated: (i) meteorological, (ii) hydrological, and (iii) agricultural droughts. Any type of drought needs to be defined in the context of local and seasonal characteristics implying that a drought should be identified rather as anomalous condition than based on an absolute threshold.

Meteorological droughts are negative anomalies in precipitation. Depending on the local characteristics, a drought can be defined as an extended period of daily precipitation amounts below a given threshold. The threshold value is defined as the minimum amount of precipitation that is needed to recharge the soil moisture content. This approach requires good knowledge

- of the local and seasonal characteristics of the soil moisture content. However, it is a useful analysis to investigate climate models' distributions of wet/dry periods which are indicative of how well suited the model is to couple to hydrological impact models. E.g., CMIP5 models have been shown to generally underestimate the number of consecutive dry days (Sillmann et al., 2013b; Cheng et al., 2016). The standardized precipitation index (SPI, McKee et al., 1993) describes local precipitation anomalies and is often used to identify Meteorological droughts. The SPI was developed as a replacement of the commonly
- used Palmer drought indices (Palmer, 1965) to better capture dry and wet anomalies. The SPI is calculated using monthly mean precipitation. Therefore, it does not account for the intensity of single precipitation events and the runoff process. Furthermore, SPI does not account for evaporation from the surface This implies, that one component of the water fluxes at the surface is lacking, which makes SPI incompatible with the concept of hydrological droughts. Evaluation of SPI from CMIP5 models shows large model biases (Ukkola et al., 2018).
- A hydrological drought occurs when low water supply effects streams, reservoirs, and groundwater levels and is usually caused by extended periods of meteorological droughts. These hydrological processes are usually not simulated with sufficient details in climate models. As a consequence, also agricultural droughts (i.e. when crops become affected by the hydrological drought) cannot be simulated properly by the models. Hydrological droughts can, however, be estimated in climate models by accounting for evapotranspiration. This allows the estimation of surface water retention. The standardized precipitation-
- evapotranspiration index (SPEI, Vicente-Serrano et al., 2010) has been developed to take into account the effect of evapotranspiration on surface water fluxes. Evapotranspiration is typically not provided by CMIP models, so SPEI often takes other inputs to estimate the it, e.g. with the Thornthwaite method based on temperature (Thornthwaite, 1948), the Hargreaves method using the monthly mean of daily minimum and maximum near-surface temperature (tasmin and tasmax) (Hargreaves,

1994), or the Penman-Monteith method using minimum and maximum temperature together with 2 m wind speed (Allen et

- al., 1994), which is estimated from the surface wind (at 10 m). However, it has been shown that the method used to derive the potential evapotranspiration has little impact on the drought statistics (Burke et al., 2006). In contrast to this finding, (Shaw and Riha, 2011) conclude that especially for future scenarios with rising temperatures potential evapotranspiration based on estimates considering temperature only can lead to an overestimation of SPEI.
- In order to assess the performance of drought characteristics in climate models, three diagnostics have been implemented into 195 the ESMValTool (v2.0): consecutive dry days, SPI and SPEI. The consecutive dry days diagnostic (recipe consecdrydays.yml) has been implemented consistently with the CDO method 'eca cdd' (Climate Data Operators, Schulzweida, 2018), and the SPI (recipe\_spei.yml) SPEI diagnostics are based the SPEI and on R-package (https://cran.rproject.org/web/packages/SPEI.pdf; Vicente-Serrano et al., 2010). The recipe *recipe spei.yml* computes the SPI and SPEI quantities for each model and summarizes the statistics of both indices as global averages in categories from "extremely
- dry" to "extremely wet", see Fig. 3 and Fig. 4. By including an estimate for evapotranspiration, the model biases are reduced, particularly for the too frequent "moderately wet" category. For SPI (Fig. 3), the bias plot shows a clear underestimation of dry and wet conditions, which are mainly compensated by too frequent moderately and extremely wet conditions. For the neutral condition category, the results differ depending on the models with a tendency towards too frequent occurrence in most models. For SPEI (Fig. 4) the bias plot indicates too frequent neutral conditions, at the expense of mainly dry and wet
- conditions. Moderate and extreme wet conditions are overestimated in practically all models, whereas moderately and extreme dry conditions show the opposite behaviour.

Using the SPI calculation described above, a recipe analysing drought events (*recipe\_martin18.yml*) has been developed. Following Martin (2018), a drought event is defined as any consecutive number of months with "extremely dry" conditions (SPI < -2). The characteristics of these events from historical and future scenario model runs (see Fig. 5) as well as from

- observational data are then compared. The characteristics investigated are frequency, length, average SPI, and the severity index following Peters (2014), which is a measure combining the length and the SPI value of a drought. Figure 5 shows an increase in the number of drought events, the severity index and to a smaller amount the duration of drought events in the RCP8.5 scenario compared to the historical model runs, especially in the subtropical areas. The results support the finding, that regions with already dry conditions are much more likely to show a higher number of drought events for the RCP 8.5
- scenario, known as the "dry gets drier and the wet gets wetter" (DDWW) paradigm (Greve et al. 2014).

# **3.2 Extreme events**

Changes in climate extremes are of utmost concern for society as the consequences of climate change will be strongly manifested in the severe impacts of extreme events, such as heatwaves and extreme precipitation, on human and natural systems. Some confidence in future projections of extreme events can be gained by evaluating the models' performance in simulating historical events against observational data and reanalysis datasets. The 27 core climate extremes indices defined

by the ETCCDI (Zhang et al., 2011) are able to capture different characteristics of temperature and precipitation extremes and are suitable for monitoring observed climate extremes, model evaluation and analysis of changes in climate extremes in future climate projections (e.g. Sillmann et al., 2013a; Sillmann et al., 2013b; Donat et al., 2013). To calculate these indices, daily values of total precipitation (pr), daily mean near-surface air temperature (tas), daily minimum near-surface air temperature (tasmin) and maximum minimum near-surface air temperature (tasmax) are required.

The *recipe\_extreme\_events.yml* calculates climate extremes indices and produces diagnostic figures for comparing model and observational extremes indices as presented in IPCC AR5 chapter 9 (Flato et al., 2013) and Sillmann et al. (2013a). The index computation is performed according to Zhang et al. (2005b). The indices are calculated from CMIP models and gridded observational/reanalysis data. Calculating the indices can take several hours up to days, depending on the number of

models/observations, length of the time periods analysed and spatial resolution of the datasets as well as the computational resources. If possible, it is recommended to run this processing step on a parallel computing system, taking advantage of the ESMValTool task-based parallelization feature (Righi et al., 2020).

There are two types of diagnostic plots that can be produced together and that reproduce the analysis shown in figure 9.37 of IPCC AR5 (Flato et al., 2013) for a given reanalysis and model dataset. The first one (see Fig. 6), shows time series providing

- a temporal comparison between the mean and spread (interquartile range) of the CMIP5 model ensemble and the individual observations for a single index. In Fig. 6, the agreement in trends between the CMIP5 models and reanalyses can be captured very well, due to the construction of the percentile-threshold based indices. Deviations from the nominal level of 10% outside the base period are mainly due to differences in the estimated trends in tasmin and tasmax of the individual models as compared to the respective reanalysis data set. In Sillmann et al. (2014) an alternative approach is described to evaluate percentile-threshold based indices in the mean.
- The second diagnostic plot (Fig. 7) shows performance metrics "portrait diagram", which compare multiple models with up to 4 different observations for multiple indices. The root-mean-square error (RMSE) between each model and each observational/reanalysis dataset is used as a measure for model performance. Figure 7 shows that the magnitude of median RMSE normalized by the spatial standard deviation of the index climatology
- in the reanalyses (RMSEstd) is generally larger for precipitation indices than for the absolute and percentile-threshold indices based on temperature with the exception of csdi and wsdi. For the temperature-based percentile-threshold indices (i.e., tx90p, tx10p, tn90p, and tn10p), the models generally perform well (except IPSL-CM5A-LR) due to their construction. This results in good agreement for the ensemble mean and medians compared to reanalysis data, whereas the root-mean-square error is too large as it is dominated by the outlier model (IPSL-CM5A-LR).
- Indices of climate extremes are a natural extension of those on the hydrological cycle discussed in Sect. 3.1 and an effort was made to make them available within the same analysis tool. As mentioned before, the ETCCDI computed by recipe\_extreme\_events.yml can be further processed by the recipe recipe\_hyint\_extreme\_events.yml. Analogous to the recipe\_hyint.yml (see also Sec. 3.1.1), it computes maps and box-averaged time series for pre-selected continental or user defined regions, computing trends and performing significance testing over the complete set of 6+27 indices. Depending on

the specific objective, the user can select the needed subset of indices. Significance testing is performed with a Student's t-test on non-null coefficients hypothesis and trend coefficients are stored together with their statistics. The recipe produces a variety of plot types for the indices, including maps and time series with their spread, trends, and summary plots of trend coefficients.

# 3.3 Impacts of climate change

# 3.3.1 Heat and cold wave duration

Heat waves are expected to become one of the greatest threats to human health in the 21st century due to projected increases in both frequency and severity (IPCC, 2013; Ouzeau et al., 2016), while the duration, intensity and frequency of cold waves are expected to decrease. It is not clear yet, however, what the impact of changes in heat and cold waves on related mortality will be, since mortality due to heat waves and cold waves inferred from historical simulations is typically overestimated. This is partly due to challenges in the correct simulation of extremes (Wang et al., 2016). In the case of heat waves in particular, models have been shown to contain biases in the 90th and 10th percentiles over the historical period (Pereira et al., 2017). However, by using a bias adjustment method based on percentiles, climate models are able to produce output which is consistent with events observed during the historical period (Ouzeau et al., 2016).

The diagnostics of the *recipe\_heatwaves\_coldwaves.yml* uses the daily maximum or minimum temperatures to estimate the relative change in heat as well as cold wave characteristics in future climates compared to a reference period. The user selects

- the model, emissions scenario, the region of interest and the reference as well as the projection periods and the percentile which will be used to compute the threshold for exceedance or non-exceedance from the reference period (a separate threshold is computed for each day of the selected season and grid point using the quantile bootstrapping method described in Zhang et al. (2005b)). Further options, which can be selected include whether to compute the frequency of exceedances or non-exceedances of extreme high or extreme low temperature events, respectively. Additionally, the minimum duration of an event to be
- classified as a heat/cold wave and the season of interest can be set. The diagnostic calculates the number of consecutive days over which temperature exceeds or does not exceed the given threshold in future climate projections. The result is presented as annual time series of the total number of heat or cold wave days for the selected season at each grid point and the average number of these days for the selected season in the future climate projections is calculated, see Fig. 8.

# 3.3.2 Combined Climate Extreme Index

High mortality rates, increases in hospital admissions as well as major economic losses are often associated with extreme events (Meehl et al., 2000; Zhang et al., 2011; Fouillet et al., 2006; Whitman et al., 1997). This emphasizes the need for monitoring and forecasting extreme events, in particular since some studies suggest that extremes are increasing in both frequency and severity with increasing anthropogenic greenhouse gases (Alexander et al., 2006; Donat et al., 2013).

The recipe recipe\_extreme\_index.yml allows a user to compute the Combined Climate Extreme Index, which is defined as a

- combination of different extreme values linked to precipitation, surface temperature and surface wind speed. This index is similar to the Climate Extremes Index (CEI; Karl et al., 1996), the modified CEI (mCEI; Gleason et al., 2008) or the Actuaries Climate Index (ACI; American Academy of Actuaries, 2018). In *recipe\_extreme\_index.yml*, the user defines the area, the reference period, the period of interest and the weights assigned for each individual component of the index. The weights allow the user to put the emphasis on the extremes that are more relevant to them and/or completely exclude non relevant ones.
- Temperature and precipitation extremes are defined in a similar fashion as in Donat et al. (2013) and are part of the larger set of extreme indices compiled by the ETCCDI (Zhang et al., 2011). The different components of the multi-metric index are
  - weight\_t90p: the number of days when the maximum temperature exceeds the 90th percentile,
  - weight\_t10p: the number of days when the minimum temperature falls below the 10th percentile,
  - weight\_Wx: the number of days when wind power (third power of wind speed) exceeds the 90th percentile,
- *weight\_cdd*: the maximum length of a dry spell, defined as the maximum number of consecutive days when the daily precipitation is below 1 mm, and
  - *weight\_rx5day:* the maximum precipitation accumulated during 5 consecutive days.

The thresholds are computed for each day in a season using a five-day running window as described in (Zhang et al., 2005a). For the calculation of the index a user-defined reference period is used for normalization and computation of the threshold

corresponding to the selected metric. This recipe creates a plot containing the time average of the components listed above for the period of interest (Fig. 9a to 9e). The recipe also computes the area-weighted average of those components and combines them into a single index using the weights and the running mean (*running\_mean* parameter) defined by the user. The output of the recipe consists of a netCDF file of the area-weighted and multi-model multi-metric index and a plot of the time series of that index over the selected period.

# 305 **3.3.3 Daily temperature range variation**

The daily temperature range (DTR) corresponds to the difference between the minimum and maximum temperature within a period of 24 hours at a given location. The usefulness of the global-average DTR has been proved using both observations and climate model simulations (Braganza et al., 2004). Changes in the mean and variability of the DTR have been shown to have a wide range of impacts on society, for example on the transmission of diseases (Lambrechts et al., 2011; Paaijmans et al.,

2010) and energy consumption (Déandreis et al., 2014).

In the energy sector, a vulnerability indicator based on the DTR has been defined to identify locations which may experience increased diurnal temperature variations in the future (Déandreis et al., 2014). Increased diurnal temperature variations put additional stress on the operational management of urban heating systems. A measure for increased diurnal temperature variations is defined as the DTR exceeding the value of the reference period by 5 K at a given location and for a given day of

315 the year. Projections of this measure are currently subject to large uncertainties as both projections of daily maximum and minimum near-surface temperature (tasmax and tasmin) in future climate projections are highly uncertain.

The recipe *recipe\_diurnal\_temperature\_index.yml* computes the mean DTR for a given reference period using historical simulations and then the number of days on which the DTR in future climate projections exceeds that of the reference period by 5K or more. The user can define both the reference and projection periods, and the region to be analysed. The output

produced by this recipe consists of a four-panel plot showing the maps of the projected mean DTR indicator for each season (see Fig. 10) and a netCDF file containing the corresponding data.

# **3.3.4 Capacity factor**

The energy sector is the largest contributor to greenhouse gas (GHG) emissions (IPCC, 2014b). Therefore, many countries have adopted mitigation strategies to increase the fraction of energy generated from renewable sources in the forthcoming

- 325 years. However, renewable energy sources like wind power and solar power rely heavily on atmospheric conditions to produce energy and are therefore exposed to risks from climate variability and long-term change in case they lead to detrimental atmospheric conditions. The relationship between wind speed and energy production by wind turbines is highly non-linear because turbines are designed to be efficient for a narrow band of wind speed conditions. Therefore, changes in the wind speed distribution can impact electricity generation and thus the revenues and economic viability of wind farms. The capacity factor
- is a normalized indicator of the suitability of wind speed conditions to produce electricity, irrespective of the size and number of installed turbines. The factor is provided for wind turbines designed for low, medium and high wind speed conditions in grouped in three different classed (IEC, 2005).

The recipe *recipe\_capacity\_factor.yml* computes the wind capacity factor for these three wind turbine classes (see Fig. 11), taking as input the daily instantaneous surface wind speed, extrapolating to the wind speed at 100 m height as described in

(Lledo et al., 2019). The user can select the region, period and season of interest. The result of the recipe is the capacity factor for each of the three turbine classes saved as netCDF file.

The output of solar photovoltaic (PV) systems depends on the time of the day, season, and weather conditions. The PV capacity factor is a measure of which fraction of the maximum possible energy is produced per grid cell. The solar power generation of a PV system mainly depends on the amount of incoming surface solar radiation but is also influenced by other atmospheric

variables that affect the efficiency of PV cells, which decreases as their temperature increases. The *recipe\_pv\_capacity\_factor.yml* computes the PV capacity factor using the daily incoming surface solar radiation and the surface temperature with a method described in Bett and Thornton (2016). The user can select temporal range, season, and region of interest. An example for is shown in Fig. 12 for ERA-Interim and five CMIP 5 models.

# **3.4 Applications for regional scales**

### 345 3.4.1 Evaluation of global climate models for selected regions

Climate or Earth system models with a fully coupled ocean are important tools to project the future evolution of the climate system in response to anthropogenic forcings, such as the increase in GHG concentrations. Despite their coarse horizontal resolutions (typically in the order of a hundred kilometres or less) these models can provide climate information at the regional scale to allow for assessing the impacts of climate change. The ability of these models to simulate regional climate is an

350 important aspect of model evaluation.

> The recipe *recipe flato13ipcc.yml* includes a subset of diagnostics and figures from the model evaluation chapter of the IPCC AR5 (Chapter 9, Flato et al., 2013), which compare surface parameters (such as temperature and precipitation) from models and observations at regional scales.

Mean seasonal cycle of precipitation and temperature is calculated over land areas within selected regions for individual

- models, the multi-model mean and observation/reanalysis data (see Fig. 13). Regional biases, including 5th, 25th, 50th, 75th and 95th percentiles of the biases, in seasonal and annual mean temperature and precipitation are evaluated for several land, polar and oceanic regions (see Figs. 14 and 15). Diagnostics allow the comparison of the multi-model mean for different projects (i.e. CMIP3, CMIP5) including information on the amplitude of the root-mean-square error. The regions used in this recipe can be irregular polygons and are defined following the IPCC Special Report on Managing the Risks of Extreme Events
- and Disasters to Advance Climate Change Adaptation (SREX) land regions (Seneviratne et al., 2012). In addition to the regions described here, the ESMValTool preprocessor can be used to run many diagnostics on distinct regions defined by latitude and longitude limits. We plan to also include regions with more complex boundaries like the CORDEX (Coordinated Regional Downscaling Experiment) regions (Gutowski et al., 2016).

Systematic biases in modelled projections (Boberg and Christensen, 2012) can be investigated by ranking models against 365 observed monthly mean temperature (see Fig. 16).

# 3.4.2 Stochastic Downscaling

The stochastic downscaling recipe is an example of how the ESMValTool (including its pre-processing functionalities) can be used to create a post-processing chain for further downscaling applications, but strictly speaking not a diagnostic.

The application of climate model projections and forecasts to impact studies at small scales, such as hydrological modelling

or ecological modelling, requires to bridge the large gap between the spatial resolution of current global and regional climate models and the scales required for a correct representation of the spatial and temporal structure of precipitation at fine-scales and of the probability of extreme precipitation events. In absence of a dynamical, physically based representation, a possible approach is the use of stochastic rainfall downscaling techniques. In particular, the Rainfall Filtered AutoRegressive Model (RainFARM; Rebora et al., 2006; D'Onofrio et al., 2014; Terzago et al., 2018) method is a weather generator which has only 375 one free parameter (which can be derived from the large scales) and which requires no further calibration. RainFARM can create ensembles of high-resolution precipitation fields from coarse scale climate model data. This method also allows quantification of uncertainties and a realistic representation of subgrid-scale variability of precipitation and of precipitation extremes, which is a crucial prerequisite for impact studies in the water sector.

The recipe *recipe\_rainfarm.yml* allows running RainFARM within the ESMValTool. Downscaled output can be produced directly from the climate model results read by the ESMValTool and exploiting its input checking, validation and pre-processing features. The recipe produces ensembles of downscaled fields (see Fig. 17) over selected regions in netCDF format, which can then be used by users for further analysis. Notice how the downscaled fields introduce fine scale precipitation structures, while still maintaining on average the original coarse-resolution precipitation. Different stochastic realizations are shown to demonstrate how an ensemble of realizations can be used to reproduce unresolved sub grid variability.

# 385 3.5 Multi-model ensemble member sub-selection

Large multi-model ensembles are a way to assess model and scenario uncertainties in future climate projections and other model experiments. However, considering constraints in the availability of computer time and human resources, not all available ensemble members can be included in most detailed climate impact studies associated to a given future scenario. Therefore, despite the importance of using an ensemble that is representative for the region and process of interest covering

- their full uncertainty range, one or few ensemble members are often rather subjectively selected depending on, for example, their availability and simplicity to access the datasets. Using more specific information about the needs of the impact study as guidance for the selection of simulations, the resulting subset can be better suited for the purpose of climate change impact research. Here, we present an efficient and flexible tool that makes better use of the ensemble by reducing its size while maintaining important ensemble characteristics.
- To find an optimal subset of significantly different model projections for a given emission scenario, a clustering algorithm is applied to the multi-model ensemble for data reduction. This technique is already used to characterize the most likely scenarios in an ensemble of weather forecasts (Ferranti and Corti, 2011; Straus et al., 2017). Similar methodologies also based on cluster analysis have been explored to select a subset from an ensemble of climate simulations (Wilcke and Barring, 2016). This approach, applied at a regional level, can also be used to identify the subset of climate model ensemble members that best represent the full range of results for further downscaling applications.
- The choice of the ensemble members is made flexible in order to meet the requirements of specific (regional) climate products and can be defined according to region and user needs. The decision of which variables are considered depends on the type and goals of the climate change impact assessment. For example, a study on future hydrological floods would require in particular changes of precipitation extreme quantiles, a study on the impact of climate change on the exploitation of ski slopes
- would require information about changes in winter temperatures and precipitation.

EnsClus (recipe *recipe\_ensclus.yml*) is a cluster analysis tool in written in Python for ensembles of climate model simulations. The tool is based on the k-means algorithm with the aim to group ensemble members by similar characteristics and to select the most representative member for each cluster. The user chooses which characteristic is used to group the ensemble members by the clustering: maximum, a given percentile (75% in the example below), mean, standard deviation or trend over the period.

- For each ensemble member this value is computed at each grid point. This results in N latitude-longitude maps, where N is the number of ensemble members. The anomalies are computed by subtracting the ensemble mean of these maps from each of the individual maps. The anomalies are therefore not computed with respect to time but to the ensemble members An Empirical Orthogonal Function (EOF) analysis is performed on these anomaly maps. For the EOF analysis, the user can set either how many Principal Components (PCs) should be calculated or the minimum percentage of the explained variance which should
- be covered. After reducing dimensionality via EOF analysis, the k-means algorithm is applied using the selected PCs (the number k of clusters needs to be defined prior to the analysis). The output of the recipe is a classification by clusters, i.e. which ensemble member belongs to which cluster and the most representative ensemble member for each cluster, defined by the member being closest to the cluster centroid. Additionally, output of the recipe includes the statistics of clustering: in the PC space, the minimum and the maximum distance between a member in a cluster and the cluster centroid (i.e. the closest and the
- farthest member), and the intra-cluster standard deviation for each cluster (i.e. compactness of the cluster). An example is shown in Fig. 18. The figure shows a clustering based on the 75th percentile of historical summer (JJA) precipitation rate for 32 CMIP5 models for the period 1900-2005. Based on the principal components explaining 80% of the variance three clusters are computed. The green cluster is the most populated with 16 ensemble members. It is mostly characterized by a positive anomaly over central-north Europe. The red cluster contains 12 ensemble members. It exhibits a negative anomaly centred
- over southern Europe and in few cases (e.g. No.12 and No.23) extending north. The third cluster (blue) includes only 4 models. It is showing a north-south dipolar precipitation anomaly, with a wetter than average Mediterranean counteracting dryer North-Europe. Ensemble members No.9, No.26 and No.19 are the "specimen" of each cluster, i.e. the model simulations that best represent the main features of that cluster. These three ensemble members can eventually be used as representative of the whole possible outcomes of the multi-model ensemble distribution associated to the 32 CMIP5 historical integrations for the
- summer precipitation rate 75th percentile over Europe. This reduces the outcomes from 32 to 3 ensemble members. The number of ensemble members of each cluster might provide a measure of the probability of occurrence of each cluster. However, the final results are sensitive to models' bias and to the metric used, as in any selection exercise.

# 4. Summary

This paper summarizes the recipes available within the ESMValTool v2.0 for the analysis of extreme events, droughts, model impact assessment, sub-selection of multi-model ensemble members e.g. for downscaling applications, as well as model evaluation on regional scales. It complements the series of papers that have been published on ESMValTool v2.0 by Righi et al. (2020) describing the technical aspects of ESMValTool v2.0, Eyring et al. (2020) presenting the new large-scale diagnostics that have been included in v2.0 since the first release in 2016 (Eyring et al., 2016), and Lauer et al. (2020) covering emergent constraints and diagnostics for the analysis of future projections from ESMs in CMIP.

- For droughts, recipes calculating the consecutive number of dry days, the SPI, and the SPEI have been newly included in ESMValTool v2.0 as well as a recipe to analyse the frequency, length, and severity of drought events based on the SPI. For further analysis of extreme events, climate extreme indices of the Expert Team on Climate Change Detection and Indices (ETCCDI) based on Zhang et al. (2011) have been included. These indices are calculated based on daily total precipitation, and the mean, minimum and maximum of the near-surface air temperature. The indices can then be plotted, used as a measure
- of model performance, and further processed to calculate index trends and their significance. For model impact assessments, recipes to analyse heat and cold wave duration, diurnal temperature variations, as well as different extreme indices are included in ESMValTool v2.0. Additional recipes compute capacity factors to analyse the impact of climate change on the wind and solar energy production.

For the analysis of ensembles of climate models, ESMValTool v2.0 provides a cluster analysis based on a k-means algorithm

where the ensemble members are divided into clusters and can be plotted along with the properties of the clusters and the most representative member of each cluster.

ESMValTool v2.0 also includes diagnostics for model evaluation on regional scales. Surface parameters such as temperature and precipitation can be evaluated for regions defined by polygons following the SPEX definitions of land regions. Additionally, the ESMValTool output can be used to be processed further by tools for stochastic downscaling, like RainFARM

which is also implemented in v2.0.

Although the recipes here are presented using CMIP5 data, ESMValTool v2.0 can be run to perform the same analysis for CMIP6 data. As an open-source project, the capabilities of the ESMValTool continue to grow with contributions from the scientific community being highly welcome. Users can analyse data using a wealth of existing recipes or join the ESMValTool development team and add new recipes and diagnostics.

# 460 **5. Code availability and data availability**

Code and data availability. ESMValTool v2.2 is released under the Apache License, Version 2.0. The latest release of ESMValTool v2.2 is publicly available on Zenodo at <a href="http://doi.org/10.5281/zenodo.4562215">http://doi.org/10.5281/zenodo.4562215</a> (Andela et al., 2021a). The source code of the ESMValCore package, which is installed as a dependency of the ESMValTool v2.2, is also publicly available on Zenodo at <a href="http://doi.org/10.5281/zenodo.4525749">http://doi.org/10.5281/zenodo.4562215</a> (Andela et al., 2021a). The source code of the ESMValCore package, which is installed as a dependency of the ESMValTool v2.2, is also publicly available on Zenodo at <a href="http://doi.org/10.5281/zenodo.4525749">http://doi.org/10.5281/zenodo.4525749</a> (Andela et al., 2021b). ESMValTool and ESMValCore are

465 developed on the GitHub repositories available at <u>https://github.com/ESMValGroup</u> (last access: 24 July 2020). CMIP5 data are available freely and publicly from the Earth System Grid Federation. Observations used in the evaluation are detailed in

the various sections of the paper and listed in Table 1. They are not distributed with ESMValTool, which is restricted to the code as open-source software.

# Author contribution

KW led the writing of the paper and with the help of LB, BKG AL, MR, MS, and ND coordinated the implementation of the diagnostics for this paper in ESMValTool v2.0. VE coordinated the ESMValTool v2.0 release. All other authors contributed to individual diagnostics for this release. All authors contributed to the text.

# Acknowledgements

- The diagnostic development of ESMValTool v2.0 for this paper was supported by different projects with different scientific focus, in particular by: (1) Copernicus Climate Change Service (C3S) "Metrics and Access to Global Indices for Climate Projections (C3S-MAGIC)" project C3S\_34a Lot 2, (2) Horizon 2020 European Union's Framework Programme for Research and Innovation under Grant Agreement No. 641816, project CRESCENDO (Coordinated Research in Earth Systems and Climate: Experiments, kNowledge, Dissemination and Outreach), (3) Helmholtz Society project "Advanced Earth System
- Model Evaluation for CMIP (EVal4CMIP)", and (4) Federal Ministry of Education and Research (BMBF) CMIP6-DICAD project. In addition, we received technical support on the ESMValTool v2.0 development from the European Union's Horizon 2020 Framework Programme for Research and Innovation "Infrastructure for the European Network for Earth System Modelling (IS-ENES3)" project under Grant Agreement No 824084. We acknowledge the World Climate Research Program's (WCRP's) Working Group on Coupled Modelling (WGCM), which is responsible for CMIP, and we thank the climate
- modelling groups for producing and making available their model output. This work used JASMIN, the UK collaborative data analysis facility as well as the DAS-5 (The Distributed ASCI Supercomputer 5) experimental supercomputer (Bal et al., 2016). The computational resources of the Deutsches Klimarechenzentrum (DKRZ, Germany) were also essential for developing and testing this new version and are kindly acknowledged.

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

Table 1. Overview of recipes implemented in ESMValTool v2.0 along with the section they are described, a brief description, the variables used,

and the diagnostic scripts included. For further details, we refer to the GitHub repository and documentation at https://docs.esmvaltool.org/.

| Recipe name                     | Section<br>(Figures)     | Description, References                                                                                                                                                                                                                                                                                                                                                                                                                                                                 | Variables<br>(Observational<br>datasets) | Diagnostic scripts                                  |
|---------------------------------|--------------------------|-----------------------------------------------------------------------------------------------------------------------------------------------------------------------------------------------------------------------------------------------------------------------------------------------------------------------------------------------------------------------------------------------------------------------------------------------------------------------------------------|------------------------------------------|-----------------------------------------------------|
| Section 3.1: Hydrological cycle |                          |                                                                                                                                                                                                                                                                                                                                                                                                                                                                                         |                                          |                                                     |
| recipe_hyint.yml                | 3.1.1<br>(Fig 1)         | Recipe for evaluating the intensity of the<br>hydroclimatic cycle, calculating a set of 6<br>indices following Giorgi et al. (2011, 2014):<br>simple precipitation intensity index (SDII),<br>maximum dry spell length (DSL) and wet<br>spell length (WSL), hydroclimatic intensity<br>index (HY-INT), which is a measure of the<br>overall behaviour of the hydroclimatic<br>cycle, and precipitation area (PA), i.e. the<br>area over which on any given day<br>precipitation occurs. | pr                                       | hyint/hyint.R                                       |
| recipe_hyint_extreme_events.yml | 3.1.1<br>(Fig 2)         | Multi-diagnostic version of hyint,<br>which allows to include ETCCDI<br>results from the extreme_events<br>diagnostics and performs joint analysis<br>of indices for hydroclimatic intensity<br>and extreme events. Giorgi et al. (2014);<br>Giorgi et al. (2011); (Sillmann et al. 2013a)                                                                                                                                                                                              | pr<br>tasmin<br>tasmax                   | hyint/hyint.R<br>extreme_events/extreme_events.R    |
| recipe_consecdrydays.yml        | 3.1.2                    | Dry day definition (precip limit<br>mm/day) and drought duration (days)<br>can be set by the user.<br>Output as netCDF files for each model<br>possible. Computed consistently with<br>the CDO method "eca_cdd" in<br>Schulzweida (2018)                                                                                                                                                                                                                                                | pr                                       | droughtindex/diag_cdd.py                            |
| recipe_spei.yml                 | 3.1.2<br>(Figs. 3,<br>4) | Global average histogram of SPI and<br>SPEI, as absolute values and as bias.<br>The calculations are based on pr for<br>both indices, but for SPEI with the<br>additional use of ta to derive                                                                                                                                                                                                                                                                                           | pr<br>(ERA-Interim)<br>ta<br>(CRUts4.01) | droughtindex/diag_spi.r<br>droughtindex/diag_spei.r |

|                                                |                        | evapotranspiration using the<br>Thornthwaite method.<br>Requires a reference dataset and<br>calculates a global cosine of latitude<br>weighted histogram for all valid grid<br>points of the reference data set.<br>Calculation of SPI and SPEI based on<br>Vicente-Serrano et al. (2010)                                                                                                                                                                                                                                                               |                                                    |                                                                                                                                                            |  |
|------------------------------------------------|------------------------|---------------------------------------------------------------------------------------------------------------------------------------------------------------------------------------------------------------------------------------------------------------------------------------------------------------------------------------------------------------------------------------------------------------------------------------------------------------------------------------------------------------------------------------------------------|----------------------------------------------------|------------------------------------------------------------------------------------------------------------------------------------------------------------|--|
| recipe_martin18grl.yml                         | 3.1.2<br>(Fig. 5)      | Computes a monthly time series of SPI,<br>based on diag_spi.r (distribution and<br>representing time scale can be set by the<br>user) and calculates drought events as<br>consecutive number of months with SPI<br>< -2. For each grid point the drought<br>characteristics (frequency, average<br>duration, and SPI as well as the severity<br>index) based on<br>Martin (2018) are calculated.<br>Differences between individual models<br>or a multi model mean and observations<br>or future scenarios and historical model<br>runs are calculated. | pr (CRU)                                           | droughtindex/diag_save_spi.R<br>droughtindex/collect_drought_func.py<br>droughtindex/collect_drought_obs_multi.py<br>droughtindex/collect_drought_model.py |  |
| Section 3.2: Extreme events                    |                        |                                                                                                                                                                                                                                                                                                                                                                                                                                                                                                                                                         |                                                    |                                                                                                                                                            |  |
| recipe_extreme_events.yml                      | 3.2<br>(Figs. 6,<br>7) | Calculate indices for monitoring<br>changes in extremes (Sillmann et al.,<br>2013a) based on daily temperature and<br>precipitation data. Produces Glecker<br>and time series plots as shown in the<br>IPCC AR5 report (Flato et al., 2013)                                                                                                                                                                                                                                                                                                             | pr<br>tas<br>tasmin<br>tasmax<br>(ERA-<br>Interim) | extreme_events/extreme_events.R                                                                                                                            |  |
| Section 3.3: Evaluation for impact assessments |                        |                                                                                                                                                                                                                                                                                                                                                                                                                                                                                                                                                         |                                                    |                                                                                                                                                            |  |
| recipe_heatwaves_coldwaves.yml                 | 3.3.1<br>(Fig. 8)      | MAGIC, time averages, differences<br>between historical simulations and a<br>future scenario, calculates the number<br>of days exceeding a given quantile for a<br>minimum number of consecutive days.<br>Watts et al. (2015)                                                                                                                                                                                                                                                                                                                           | Tasmin<br>Tasmax                                   | magic_bsc/extreme_spells.r                                                                                                                                 |  |

| recipe_extreme_index.yml              | 3.3.2<br>(Fig. 9)                     | MAGIC, computes time series of the<br>number of several extreme events:<br>heatwave,<br>cold wave, heavy precipitation,<br>drought, and high wind.<br>Karl et al. (1996); Gleason et al. (2008);<br>American Academy of Actuaries<br>(2018)                                                                                                                                                                                                                      | Tasmin<br>tasmax<br>pr<br>scfWind      | magic_bsc/extreme_index.r                                                                                                                                |
|---------------------------------------|---------------------------------------|------------------------------------------------------------------------------------------------------------------------------------------------------------------------------------------------------------------------------------------------------------------------------------------------------------------------------------------------------------------------------------------------------------------------------------------------------------------|----------------------------------------|----------------------------------------------------------------------------------------------------------------------------------------------------------|
| recipe_diurnal_temperature_index.yml  | 3.3.3<br>(Fig. 10)                    | MAGIC, time averages, difference<br>between historical and future scenario,<br>computes the dates where the DTR<br>exceeds a threshold,<br>Déandreis et al. (2014)                                                                                                                                                                                                                                                                                               | tasmin<br>tasmax                       | magic_bsc/diurnal_temp_index.r                                                                                                                           |
| recipe_capacity_factor.yml            | 3.3.4<br>(Fig. 11)                    | MAGIC, calculates the wind power<br>capacity factor,<br>Lledo et al. (2019)                                                                                                                                                                                                                                                                                                                                                                                      | scfWind                                | magic_bsc/capacity_factor.r                                                                                                                              |
| recipe_pv_capacity_factor.yml         | 3.3.4<br>(Fig. 12)                    | Photo voltaic capacity factor.                                                                                                                                                                                                                                                                                                                                                                                                                                   | tasmax<br>rsds<br>(ERA-Interim)        | pv_capacityfactor/pv_capacity_factor.R                                                                                                                   |
| Section 3.4: Regional model evaluatio | n                                     |                                                                                                                                                                                                                                                                                                                                                                                                                                                                  |                                        |                                                                                                                                                          |
| recipe_flato13ipcc.yml                | 3.4.1<br>(Figs.<br>13, 14,<br>14, 16) | Figures similar to figures of the IPCC<br>AR5 (Flato et al., 2013).<br>Fig. 13: Seasonal cycle over land within<br>defined regions (like Fig. 9.38)<br>Fig 14: Downscaling: Seasonal bias<br>box plot within defined regions (like<br>Fig. 9.39)<br>Fig. 15: Downscaling: Seasonal bias<br>box plot within defined polar and ocean<br>regions (like Fig. 9.40)<br>Fig. 16: Downscaling: observations<br>versus models within defined regions<br>(like Fig. 9.41) | tas (ERA-<br>Interim, CRU)<br>pr (CRU) | regional_downscaling/Figure9.38.ncl<br>regional_downscaling/Figure9.38.ncl<br>regional_downscaling/Figure9.38.ncl<br>regional_downscaling/Figure9.38.ncl |
| recipe_rainfarm.yml                   | 3.4.2<br>(Fig. 17)                    | MAGIC, Stochastic spatial<br>donwscaling of daily precipitation<br>using the RainFARM method (Rebora<br>et al. 2006; D'Onofrio et al. 2014).                                                                                                                                                                                                                                                                                                                     | pr                                     | rainfarm/rainfarm.R                                                                                                                                      |

|                                                        |               | Allows calculation of climatological<br>weights to take into account the effect<br>of orography following Terzago et al.<br>(2018). Produces ensembles of<br>downscaled precipitation fields in<br>netCDF format. No plots are produced.                                     |           |                    |  |  |
|--------------------------------------------------------|---------------|------------------------------------------------------------------------------------------------------------------------------------------------------------------------------------------------------------------------------------------------------------------------------|-----------|--------------------|--|--|
| Section 3.5: Multi-model ensemble member sub-selection |               |                                                                                                                                                                                                                                                                              |           |                    |  |  |
| recipe_ensclus.yml                                     | 3.5 (Fig. 18) | Cluster analysis tool for ensembles of<br>climate model simulations. EnsClus<br>groups ensemble members according to<br>similar characteristics (based on the k-<br>means algorithm) and selects the most<br>representative member for each cluster<br>(Straus et al., 2007) | Pr<br>tas | ensclus/ensclus.py |  |  |

HY-INT: World-EC-EARTH 2006-2099

Figure 1: Mean hydroclimatic intensity index (i.e., a combination of precipitation intensity and dry spell length normalized compared to a reference period) over the years 2006-2099, for the EC-EARTH model RCP 8.5 projection. The historical years 1976-2005 were
used as the reference period. The figure is an example of a large number of different plots which can be produced with *recipe\_hyint.yml*, similar to (Giorgi et al., 2014). For details see Section 3.1.1.

PA trend (1976-2099)