# Peer review of "Earth System Model Evaluation Tool (ESMValTool) v2.0 – diagnostics for extreme events, regional and impact evaluation and analysis of Earth system models in CMIP"

_Geoscientific Model Development, 2020_

## Referee Comment (RC1) · Anonymous Referee #1 · 29 Oct 2020

Overall: This is an overview of what looks like a very useful tool for climate model data analysis. I am not involved in CMIP, but I think this tool is going to be useful beyond CMIP. A paper discussing what the tool is about what would be helpful is making awareness of the tool's availability.

I have mostly minor comments and some additional comments about how the tool can be improved.

Comments:

Line 142: "The linear model (lm) function of R is used to calculate trends." One possible alternative is to use Generalised Linear Model (GLM; Nelder and Wedderburn 1972) of R (function glm) instead as GLMs are more flexible (with standard linear regression being part of this approach). The fitting of indices that their values do not follow normal distribution would be made more flexible and easier. It should be a fairly straightforward change to the R code as all R regression modules more or less follow the same standard.

Section 3.3.3: I would think another related metric would be the annual temperature range (warm season Tmax vs cold season Tmin) could be quite useful along with DTR. If annual temperature range is widening, it may also imply energy use be expected to increase (akin to DTR getting larger).

Lines 353-354: Are the CORDEX regions included part of this package? I think doing so will make them more useful to compare with regional climate model results.

Section 3.5: More a general comment – this will be a very useful tool in the future to just to avoid data and information overload, considering the volume of multi-model and multi-ensemble data will be involved in future MIPs.

Technical comments:

Line 162: May be better to say "Meteorological droughts are negative anomalies in precipitation." instead.

Line 173: ". . . which makes SPI incompatible with the concept of hydrological droughts."

Line 179: "This allows the estimation of surface water retention."

Line 181: "Evapotranspiration is typically not calculated by climate models." Climate model does output them as part of the land surface model output, but how that is computed are simplistic as in being diagnosed from the variables the authors are mentioning (i.e. surface T and wind). Hence, I am not sure "calculate" is the right word here. Perhaps, "is not prognostic" "is diagnosed simplistically" would be more appropriate.

Lines 224-225: It may be advisable to drop the "computationally demanding" from the sentence as a few hours or days are still short comparing to the wall clock time needed to run the CMIP models.

Figure 6, values near year 2000: The ensemble spread around year 2000 is outside of the ensemble mean. Please check what causes this.

Figure 7, lines 241: For the sake clarity and easy viewing of the figure, indicate which of the 3 indices are for precipitation (the top 3 ones).

Lines 256-261: Can you be more specific what the extreme temperature biases are? Do you mean non-bias corrected data has a lot more temperature extremes than in observations (which would be consistent what mortality rates estimates are too high)? If yes, state so directly.

Figure 9: The font size of the titles for each panel are small, and one can tell the dpi of the image is quite low (which makes the titles even harder to read). I think the dpi issue can be addressed by outputting the figure as a png (or other reasonable lossless format) or pdf.

References:

Nelder JA, Wedderburn RWM (1972) Generalized Linear Models. Journal of the Royal Statistical Society Series A (General) 135:370–384. https://doi.org/10.2307/2344614

---

## Referee Comment (RC2) · Anonymous Referee #2 · 23 Nov 2020

Overall comment:

The paper gives an overview on the new implemented data analysis and diagnostic recipes in the version 2 of ESMValTool. The prerequisite for using the open source tool is that the data not only follow the CF conventions for the file structures in netCDF but also the CMOR standard of CMIP is applied. To broaden the usability of the tool, it would be good to base the work on the given standard_names of variables and using an internal dictionary to translate variable names to the CMOR standard. That would avoid reformatting of used datasets depending on the software used (in his case

ESMValTool).

The quality of the paper in terms of structure and language is high but sometimes there is a lack of correctness and attention, which leads to several minor comments. The high number of comments leads to the overall judgement 'major revisions'.

The comments are ordered by the occurrence in the text and where the comment relates not only to the text but to the figure as well the figure-comment is brought forward.

Content:

P4L96: "Observations used in the evaluation are described in the following section' – this is not the case (describing is perhaps not necessary but mentioning them)

P5L136: "...hydroclimatic intensity index ...which is a measure of the overall behavior of the hydroclimatic cycle"– this is not true; it is a measure of change of hydroclimatic intensity. As the hydroclimatic intensity itself is normalized, the values are not comparable in space.

P28Figure1: The figure shows the mean change in hydroclimatic intensity and not the 'mean hydroclimatic intensity'.

The measure is an indicator for changes in time compared to the normal period, in this case 1976-2005. Therefore, it is astonishing that the authors give the mean change for the period 1976-2099. Please, explain your intension.

P8L228: Figure 7 is much more complex than Figure 6 – therefore it is not good to throw them together and explaining them a bit than coming back to Figure 6 and afterwards again explaining Figure 7.

P8L231(and Figure7) To start the discussion of Figure 7 it is misleading to use "The root-mean-square error..." as it is very import for understanding the colors and signs that you are talking about a normalized root-mean-square error – normalized by the

**GMDD**

[Figure]

root-mean-square error of the ensemble median. At the Figure caption (L724) is written "median RMSE normalized by the spatial standard deviation of the index climatology in the reanalyses (RMSEstd)" whereas the text (L238) gives "multi-model median error on a global scale (i.e. RMSEstd)" here the wording should be exact. "Figure 7 shows that the magnitude of the multi-model median error on a global scale (i.e. RMSEstd) is generally larger for precipitation indices than for the absolute and percentile-threshold indices based on temperature with the exception of csdi and wsdi." – it is not clear why the authors praise percentile-threshold indices as the RMSEstd seem to be even worse than the precipitation values.

P33L724: "...the ensemble mean error..." – "error" this is too short for readers not familiar with (Sillmann,2013a)

P9L271: "The result is presented as annual time series of the total number of heat or cold wave days for..." – this is not the case – in Figure 8 "summer days" are shown without any restrictions regarding the sequence of days, so the 'wave' aspect is excluded.

P10L298: It would be nice to get an example for the combined index as Fig. 9f.

P10L304: (Déandreis et al., 2014) is missing in reference list. And the document references on the ESMValTool web site (https://docs.esmvaltool.org/en/latest/recipes /recipe_diurnal_temperature_index.html?highlight=D%C3%A9andreis #diurnal-temperature-range) does not exist anymore: Déandreis C. (IPSL), Bra-connot P. (IPSL), Planton S. (CNRMGAME). Study performed for the DALKIA company. http://secif.ipsl.fr/images/SECIF/documents/ Communica-tion/fiche_invulnerable/RC_indicateur_EN.pdf

P10L307: Please, check "The measure is defined as the DTR exceeding 5°C at a given location and for a given day of the year" the suspicion suggests that "...as the DTR exceeding the value of the reference period by 5 K..."

P11L328: "daily instantaneous" - Lledo is using 6 hourly instantaneous wind speed data, which is courageous enough. Taking daily means results in strong underestimation of highest wind speeds and strongly disturbs the effect of the power curve but taking one instantaneous value is a strange idea and has to be justified.

P11L337: The daily cycle of surface incoming solar radiation is not a strength of climate models – therefore it would be very interesting to see a comparison with observed data.

P14L413: The period is given with "over 1900-2005" – please, correct.

Technical comments:

General:

- Check the space between two citations, it is sometimes missing (...Teixeira et al., 2014;Waliser et al., 2020)

- Check the space between number and unit, it should be 2 m instead of 2m (e.g. L184)

- The physical unit of temperature difference is Kelvin. Please, change °C and degree to K where necessary.

- Check reference list for unintended line breaks

P3L69 (Straus et al. 2007) is missing in reference list

P3L76 (Covey et al. 2003) is missing in reference list

P7L191 (Schulzweida, 2018) is missing in reference list

P11L326 (IEC, 2005) is missing in reference list

P13L388 (Ferranti and Corti, 2011) is missing in reference list

P9L281 'American Academy of Actuaries' should be the entry name in the reference list (instead of Actuaries Climate Index)

P17L513 'Impact du changement climatique sur la gestion des réseaux de chaleur.'is

not in the text

Figures:

P29Figure2: The headlines of the subplots give 'trend 1976-2099', whereas meant is changes in 2006-2099 with respect to 1975-2005.

P31Figure5: The plot title give "of Event" – could you change to "drought Events" to make obvious what is meant?

P31L705 Please, correct "historic (1950-2100)".

P34Figure8: "a)" and "b)" should be added to the figure. "Average number of summer days" should be replaced by "Average number of yearly summer days". 80th quantile should be 80th percentile.

The colorbar should be adjusted to the 'normal' number of days - for the exceeding of the 80th percentile 18 days out of 92: green colors for less than 18 days and yellow to red colors for more than 18 days.

P35Figure9: All units are given with 'cm', why?

P36L738-739 d) and e) are mixed and "maximum" is missing: "d) drought and e) maximum precipitation".

P34-36Figure8-10: The relation of x and y dimension of 2D-Figs 8-10 should be more realistic.

P37Figure11: Please, reduce the accuracy of the colorbar and reduce its size appropriate to the plots.

P38L777: Add the colors of observations: "as ERA-Interim (yellow line) and CRU (black line) data shown".

P39Figure13: Consider a legend for the regions abbreviations beneath the annual plot.

P40Figure14: The technical quality is poor.

**[GMDD](https://www.geoscientific-model-development.net/)**
P41Figure15: The technical quality is poor. Axis titles are insufficient (variable + unit missing).

P42Figure16 Does it make sense to give to exact date of the example - it is not comparable to reality/observed value. A hint that it is an artificial date would help.

P43Figure17: 1900-2005

Tables:

Table1 is hard to read as the column for 'Description' is rather narrow, consider landscape format if is possible and broaden the description-column.

Language/Spelling

P11L317: Please, consider rephrasing "is the single biggest contributor".

P11L326: ".."

P13L400: "example" instead of "examples"

P15L451: Do not use small caps but write "Version 2.0"

---

## Author Comment (AC1) · 8 Mar 2021

We thank referee 1 for the time and effort spent on reading the paper and providing helpful comments. A point wise reply is given below, with the original comments in italics.

*Overall: This is an overview of what looks like a very useful tool for climate model data analysis. I am not involved in CMIP, but I think this tool is going to be useful beyond CMIP. A paper discussing what the tool is about what would be helpful is making*

[Figure]

*awareness of the tool's availability. I have mostly minor comments and some additional comments about how the tool can be improved.*

We appreciate the suggestions for new metrics. The ESMValTool is continuously developed further and we welcome any technical or scientific contribution (on Github https://github.com/ESMValGroup). Unfortunately, we cannot include all changes suggested at this stage, but we provide below details about which software changes are planned or done in this respect, as well as the answers to the all other comments.

**Comments:**

*Line 142: "The linear model (lm) function of R is used to calculate trends." One possible alternative is to use Generalised Linear Model (GLM; Nelder and Wedderburn 1972) of R (function glm) instead as GLMs are more flexible (with standard linear regression being part of this approach). The fitting of indices that their values do not follow normal distribution would be made more flexible and easier. It should be a fairly straightforward change to the R code as all R regression modules more or less follow the same standard.*

Thank you for the suggestion. In the current implementation we have adopted for now only a simple linear trend line, as commonly used in most climate studies. Indeed a GLM approach could be useful, as you say, particularly for analysing indices with a non-normal distribution and we will consider implementing this functionality in future versions of the code. Please notice that, since ESMValTool produces raw netcdf files with the indices as a function of time, in addition to the plots, this type of more sophisticated analysis can already be performed separately by the user using external tools if they wish.

*Section 3.3.3: I would think another related metric would be the annual temperature range (warm season Tmax vs cold season Tmin) could be quite useful along with DTR. If annual temperature range is widening, it may also imply energy use be expected to increase (akin to DTR getting larger).*

[Figure]

There is a function of the ESMValTool preprocessor, which can provide these data, documented at: 'https://docs.esmvaltool.org/projects/ESMValCore/en/latest/api/esmvalcore.preprocessor. html?highlight=preprocessoresmvalcore.preprocessor.amplitude', recipe_wenzel16nat.yml described in Lauer et al., 2020 uses this function to plot the annual cycle of $CO_2$. However, there is no recipe or diagnostic to exploit this function for the annual temperature range, yet, but we will consider an implementation in future releases.

*Lines 353-354: Are the CORDEX regions included part of this package? I think doing so will make them more useful to compare with regional climate model results.*

It is planned to include the CORDEX regions in the ESMVal-Tool, and the development in this regards has already started, see https://github.com/ESMValGroup/ESMValCore/pull/184 To highlight this in the paper we will included the following: "In addition to the regions described here, the ESMValTool preprocessor can be used to run many diagnostics on distinct regions defined by latitude and longitude limits. We plan to also include regions with more compex boundaries like the CORDEX (Coordinated Regional Downscaling Experiment) regions (Gutowski et al., 2016)."

*Section 3.5: More a general comment – this will be a very useful tool in the future to just to avoid data and information overload, considering the volume of multi-model and multi-ensemble data will be involved in future MIPs.*

Thanks, there is a lot of effort put into optimizing the data handling, for example by using lazy data evaluation with the Dask python package (https://dask.org/, see also Righi et al. 2020).

**Technical comments:**

*Line 162: May be better to say "Meteorological droughts are negative anomalies in*

*precipitation." instead.*

Changed

*Line 173: ". . . which makes SPI incompatible with the concept of hydrological droughts."*

Changed

*Line 179: "This allows the estimation of surface water retention."*

Changed

*Line 181: "Evapotranspiration is typically not calculated by climate models." Climate model does output them as part of the land surface model output, but how that is computed are simplistic as in being diagnosed from the variables the authors are mentioning (i.e. surface T and wind). Hence, I am not sure "calculate" is the right word here. Perhaps, "is not prognostic" "is diagnosed simplistically" would be more appropriate.*

We changed the corresponding sentence to: "Evapotranspiration is typically not provided by CMIP models, ..."

*Lines 224-225: It may be advisable to drop the "computationally demanding" from the sentence as a few hours or days are still short comparing to the wall clock time needed to run the CMIP models.*

We dropped "computationally demanding" and changed the sentence to: "Calculating the indices can take several hours up to days, depending on the number of models/observations, length of the time periods analysed and spatial resolution of the datasets as well as the computational resources."

*Figure 6, values near year 2000: The ensemble spread around year 2000 is outside of the ensemble mean. Please check what causes this.*

The shading does not display the spread around the mean but the area between the

[Figure]

*Figure 7, lines 241: For the sake clarity and easy viewing of the figure, indicate which of the 3 indices are for precipitation (the top 3 ones).*

We will mark the indices for precipitation.

*Lines 256-261: Can you be more specific what the extreme temperature biases are? Do you mean non-bias corrected data has a lot more temperature extremes than in observations (which would be consistent what mortality rates estimates are too high)? If yes, state so directly.*

Ouzeau et al. (2016), do not give details on the temperature biases. Nevertheless, bias corrections derived from comparing historical model experiments with reanalysis data applying a quantile-mapping technique can increase the confidence in future simulations.

*Figure 9: The font size of the titles for each panel are small, and one can tell the dpi of the image is quite low (which makes the titles even harder to read). I think the dpi issue can be addressed by outputting the figure as a png (or other reasonable lossless format) or pdf.*

We will increase the resolution and font size of the figure.

**References:**

*Nelder JA, Wedderburn RWM (1972) Generalized Linear Models. Journal of the Royal Statistical Society Series A (General) 135:370–384. https://doi.org/10.2307/2344614*

**References in the answers:**

Gutowski, W. J., Giorgi, F., Timbal, B., Frigon, A., Jacob, D., Kang, H. S., Raghavan, K., Lee, B., Lennard, C., Nikulin, G., O'Rourke, E., Rixen, M., Solman, S., Stephenson, T., and Tangang, F.: WCRP COordinated Regional Downscaling EXperiment (CORDEX): a diagnostic MIP for CMIP6, Geosci Model Dev, 9, 4087-4095, 10.5194/gmd-9-4087-

2016, 2016.

Lauer, A., Eyring, V., Bellprat, O., Bock, L., Gier, B. K., Hunter, A., Lorenz, R., Pérez-Zanón, N., Righi, M., Schlund, M., Senftleben, D., Weigel, K., and Zechlau, S.: Earth System Model Evaluation Tool (ESMValTool) v2.0 – diagnostics for emergent constraints and future projections from Earth system models in CMIP, Geosci. Model Dev. Discuss., accepted, 2020, 1-47, 10.5194/gmd-2020-60, 2020.

Ouzeau, G., Soubeyroux, J. M., Schneider, M., Vautard, R., and Planton, S.: Heat waves analysis over France in present and future climate: Application of a new method on the EURO-CORDEX ensemble, Climate Services, 4, 1-12, 10.1016/j.cliser.2016.09.002, 2016.

Righi, M., Andela, B., Eyring, V., Lauer, A., Predoi, V., Schlund, M., Vegas-Regidor, J., Bock, L., Brötz, B., de Mora, L., Diblen, F., Dreyer, L., Drost, N., Earnshaw, P., Hassler, B., Koldunov, N., Little, B., Loosveldt Tomas, S., and Zimmermann, K.: Earth System Model Evaluation Tool (ESMValTool) v2.0 – technical overview, Geosci. Model Dev., 13, 1179-1199, 10.5194/gmd-13-1179-2020, 2020.

---

## Author Comment (AC2) · 8 Mar 2021

We thank referee 2 for the time and effort spent on reading the paper and providing helpful comments. A point wise reply is given below, with the original comments in italics.

**Overall comment:**

*The paper gives an overview on the new implemented data analysis and diagnostic recipes in the version 2 of ESMValTool. The prerequisite for using the open source*

[Figure]

*tool is that the data not only follow the CF conventions for the file structures in netCDF but also the CMOR standard of CMIP is applied. To broaden the usability of the tool, it would be good to base the work on the given standard_names of variables and using an internal dictionary to translate variable names to the CMOR standard. That would avoid reformatting of used datasets depending on the software used (in his case ESMValTool).*

We are aware that the reformatting of external datasets (like observations) to the CMOR standard is a drawback and we are working to make this more flexible in future versions of the ESMValTool. A prototype is already implemented for the ERA5 reanalysis data, for which a cmorization-on-the-fly is implemented, which works on the ERA5 netCDF data directly and does not require prior reformatting. There is also work done to implement other standards then CMOR, e.g. through the CIS Toolbox (Community Intercomparison Suite, https://www.ecmwf.int/en/elibrary/17846-community-intercomparison-suite-cis-open-source-toolbox), see also https://github.com/ESMValGroup/ESMValTool/pull/1656

*The quality of the paper in terms of structure and language is high but sometimes there is a lack of correctness and attention, which leads to several minor comments. The high number of comments leads to the overall judgement 'major revisions'. The comments are ordered by the occurrence in the text and where the comment relates not only to the text but to the figure as well the figure-comment is brought forward.*

**Content:**

*P4L96: "Observations used in the evaluation are described in the following section' – this is not the case (describing is perhaps not necessary but mentioning them)*

We changed the text to: "For the recipes described here, European Centre for Medium-Range Weather Forecasts (ECMWF) ERA-Interim and Climatic Research Unit (CRU) reanalysis data are used for the evaluation of the model results. Table 1 lists these data in case they are used for a recipe. These datasets should be seen as examples

as they can easily be replaced by other reanalysis or observational datasets. Reformatting scripts with downloading instructions are provided with the ESMValTool v2.0 to convert many observational datasets to the CMOR standard. A list of observational datasets available can be found in Righi et al. (2020) and in the user's guide at https://docs.esmvaltool.org/en/latest/input.htmlsupported-datasets", where it is updated for newly included datasets. For ECMWF ERA5 a "cmorization-on-the-fly" is implemented, which works on the ERA5 netCDF data directly and does not require prior reformatting."

*P5L136: ". . .hydroclimatic intensity index . . .which is a measure of the overall behavior of the hydroclimatic cycle"– this is not true; it is a measure of change of hydroclimatic intensity. As the hydroclimatic intensity itself is normalized, the values are not comparable in space.*

Indeed "overall" could be misleading because it could be read as "global", while clearly the index, as pointed out, is local in space. We thank the reviewer for pointing out this slight inaccuracy. As for HY-Int being a measure of change: As it is defined, HY-Int is a measure of the intensity of the hydroclimatic cycle, expressed in units normalized compared to a reference period, so that indeed its value can give indications on the change compared to the reference period. To clarify, we changed the sentence to: "....hydroclimatic intensity index....which is a measure of the intensity of the hydroclimatic cycle compared to a reference period (Giorgi et al, 2011) ...."

*P28Figure1: The figure shows the mean change in hydroclimatic intensity and not the 'mean hydroclimatic intensity'. The measure is an indicator for changes in time compared to the normal period, in this case 1976-2005. Therefore, it is astonishing that the authors give the mean change for the period 1976-2099. Please, explain your intension.*

The reviewer is right that the figure does show the "mean hydroclimatic intensity index" and not the "mean hydroclimatic intensity". In fact the figure shows the climatological

average of the HY-INT index in a given period. The index is normalized compared to a reference period, so indirectly (as discussed in the previous reply) it provides a measure of change. We added "index" after "hydroclimatic intensity" and a better summary of the definition of the index in the caption to clarify this.

Fig. 1 was indeed computed as the climatological average of the HY-INT index in the period 2006-2099, excluding the reference period. The figure title and the caption were wrongly referring instead to 1976-2099 and we now corrected them (please notice that the figure itself did not change). the averaging period is indeed 2006-2099 and not 1976-2099 as stated in the original version.

Accordingly, we changed the figure caption to the following: Figure 1: "Mean hydroclimatic intensity index (i.e., a combination of precipitation intensity and dry spell length normalized compared to a reference period) over the years 2006-2099, for the EC-EARTH model RCP 8.5 projection. The historical years 1976-2005 were used as the reference period. The figure is an example of a large number of different plots which can be produced with recipe_hyint.yml, similar to (Giorgi et al., 2014). For details see Section 3.1.1."

*P8L228: Figure 7 is much more complex than Figure 6 – therefore it is not good to throw them together and explaining them a bit than coming back to Figure 6 and afterwards again explaining Figure 7.*

To avoid confusion in the description of Figure 6 and 7, we changed this paragraph to: "There are two types of diagnostic plots that can be produced together and that reproduce the analysis shown in figure 9.37 of IPCC AR5 (Flato et al., 2013) for a given reanalysis and model dataset. The first one (see Fig. 6), shows time series providing a temporal comparison between the mean and spread (interquartile range) of the CMIP5 model ensemble and the individual observations for a single index. In Fig. 6, the agreement in trends between the CMIP5 models and reanalyses can be captured very well, due to the construction of the percentile-threshold based indices.

Deviations from the nominal level of 10

The second diagnostic plot (Fig. 7) shows performance metrics "portrait diagram", which compare multiple models with up to 4 different observations for multiple indices. The root-mean-square error (RMSE) between each model and each observational/reanalysis dataset is used as a measure for model performance. Figure 7 shows that the magnitude ... "

*P8L231(and Figure7) To start the discussion of Figure 7 it is misleading to use "The root-mean-square error. . ." as it is very import for understanding the colors and signs that you are talking about a normalized root-mean-square error – normalized by the root-mean-square error of the ensemble median. At the Figure caption (L724) is written "median RMSE normalized by the spatial standard deviation of the index climatology in the reanalyses (RMSEstd)" whereas the text (L238) gives "multi-model median error on a global scale (i.e. RMSEstd)" here the wording should be exact. "Figure 7 shows that the magnitude of the multi-model median error on a global scale (i.e. RMSEstd) is generally larger for precipitation indices than for the absolute and percentile-threshold indices based on temperature with the exception of csdi and wsdi." – it is not clear why the authors praise percentile-threshold indices as the RMSEstd seem to be even worse than the precipitation values.*

We changed the text at L238 to be consistent with the figure caption to: "Figure 7 shows that the magnitude of median RMSE normalized by the spatial standard deviation of the index climatology in the reanalyses (RMSEstd) is generally larger for precipitation indices than for the absolute and percentile-threshold indices based on temperature with the exception of csdi and wsdi." This sentence does not state a difference between percentile and non percentile indices but between precipitation and temperature indices.

*P33L724: ". . .the ensemble mean error. . ." – "error" this is too short for readers not familiar with (Sillmann,2013a)*

[Figure]

We changed the sentence to: "Blue (red) colours indicate that a model performs better (worse) than the median of all model results when compared to the respective reanalysis dataset."

*P9L271: "The result is presented as annual time series of the total number of heat or cold wave days for. . ." – this is not the case – in Figure 8 "summer days" are shown without any restrictions regarding the sequence of days, so the 'wave' aspect is excluded.*

For the recipe_heatwaves_coldwaves.yml the user can set a minimum duration of consecutive days, hence the wave aspect is included. However, this is not described detailed enough in the caption of Figure 8, hence we added: "The minimum duration of a heatwave event can be chosen in the recipe and is set to 5 days here."

*P10L298: It would be nice to get an example for the combined index as Fig. 9f.*

As suggested, we added a time series of the combined index as figure 9f.

*P10L304: (Déandreis et al., 2014) is missing in reference list. And the document references on the ESMValTool web site (https://docs.esmvaltool.org/en/latest/recipes /recipe_diurnal_temperature_index.html?highlight=Ddiurnal-temperature-*

*range) does not exist anymore: Déandreis C. (IPSL), Braconnot P. (IPSL), Planton S. (CNRMGAME). Study performed for the DALKIA company. http://secif.ipsl.fr/images/SECIF/documents/ Communication/fiche_invulnerable/RC_indicateur_EN.pdf*

We added the reference, it is now found under: https://docplayer.fr/9496504-Impact-du-changement-climatique-sur-la-gestion-des-reseaux-de-chaleur.html. We will also update the ESMValTool documentation.

*P10L307: Please, check "The measure is defined as the DTR exceeding 5°C at a given location and for a given day of the year" the suspicion suggests that ". . .as the DTR exceeding the value of the reference period by 5 K. . ."*

We changed this paragraph to: "In the energy sector, a vulnerability indicator based on the DTR has been defined to identify locations which may experience increased diurnal temperature variations in the future (Déandreis et al., 2014). Increased diurnal temperature variations put additional stress on the operational management of urban heating systems. A measure for increased diurnal temperature variations is defined as the DTR exceeding the value of the reference period by 5 K at a given location and for a given day of the year."

We also used 5 K in the following paragraph instead of 5°C

*P11L328: "daily instantaneous" - Lledo is using 6 hourly instantaneous wind speed data, which is courageous enough. Taking daily means results in strong underestimation of highest wind speeds and strongly disturbs the effect of the power curve but taking one instantaneous value is a strange idea and has to be justified.*

This issue is discussed in depth in Lledó et al. (2019) section 3.2. Using daily means would in most cases be worse than using daily instantaneous values, if the sample of days is large enough. Keep in mind that "... six-hourly instantaneous values from models are not directly comparable to six-hourly instantaneous samples from site observations". Of course, locations with a strong diurnal cycle will experience relevant biases because only data at a single time of day is used, but generally, authors believe this is the best solution given the data available.

*P11L337: The daily cycle of surface incoming solar radiation is not a strength of climate models – therefore it would be very interesting to see a comparison with observed data.*

As suggested, included a figure to show the results for the photovoltaic capacity factor for some CMIP5 modles and ERA-Interim data.

*P14L413: The period is given with "over 1900-2005" – please, correct.*

Changed to "for the period 1900-2005."

**Technical comments:**

[Figure]

**General:**

*- Check the space between two citations, it is sometimes missing (. . .Teixeira et al., 2014;Waliser et al., 2020)*

We added the spaces where they were missing between multiple references.

*- Check the space between number and unit, it should be 2 m instead of 2m (e.g. L184)*

We added the spaces.

*- The physical unit of temperature difference is Kelvin. Please, change âŮę C and degree to K where necessary.*

For some applications, °C is more widely used than K. The default in the ESMValTool is K, but for several diagnostics, e.g. the ones related to the ETCCDI Indices °C is used because this unis is used in the original publication for the metric (Zhang et al. 2011). In these cases we would prefer to keep °C.

*- Check reference list for unintended line breaks*

We removed the unintended line breaks.

*P3L69 (Straus et al. 2007) is missing in reference list*

Added

*P3L76 (Covey et al. 2003) is missing in reference list*

Added

*P7L191 (Schulzweida, 2018) is missing in reference list*

Added

*P11L326 (IEC, 2005) is missing in reference list*

Added

*P13L388 (Ferranti and Corti, 2011) is missing in reference list*

Added

*P9L281 'American Academy of Actuaries' should be the entry name in the reference list (instead of Actuaries Climate Index)*

Changed

*P17L513 'Impact du changement climatique sur la gestion des réseaux de chaleur.'is*

Added, see above.

**Figures:**

*P29Figure2: The headlines of the subplots give 'trend 1976-2099', whereas meant is changes in 2006-2099 with respect to 1975-2005.*

The caption of Fig 2 was referring to trends in the period 2006-2099, while actually the trends in the period 1976-2099 are reported (as already indicated by the figure titles). We fixed the caption of Fig 2 and we apologize for the confusion.

*P31Figure5: The plot title give "of Event" – could you change to "drought Events" to make obvious what is meant?*

We changed the title.

*P31L705 Please, correct "historic (1950-2100)".*

We corrected the text: "historic (1950 to 2000)".

*P34Figure8: "a)" and "b)" should be added to the figure. "Average number of summer days" should be replaced by "Average number of yearly summer days". 80th quantile should be 80th percentile. The colorbar should be adjusted to the 'normal' number of days - for the exceeding of the 80th percentile 18 days out of 92: green colors for less than 18 days and yellow to red colors for more than 18 days.*

We added a) and b) and changed the text to "Average annual number of summer days..." and "percentile" instead of "quantile". Although it would be possible to adjust the colors for this example, we would prefer to keep and show the default color bar used be the ESMValTool for this recipe, which should be applicable for different regions and time periods.

*P35Figure9: All units are given with 'cm', why?*

The indices are dimensionless, "cm" was removed.

*P36L738-739 d) and e) are mixed and "maximum" is missing: "d) drought and e) maximum precipitation".*

We corrected the caption.

*P34-36Figure8-10: The relation of x and y dimension of 2D-Figs 8-10 should be more realistic.*

Figure 8-10 are replaced and the dimensions are adjusted.

*P37Figure11: Please, reduce the accuracy of the colorbar and reduce its size appropriate to the plots.*

The color bar for Figure 11 was changed.

*P38L777: Add the colors of observations: "as ERA-Interim (yellow line) and CRU (black line) data shown".*

*P39Figure13: Consider a legend for the regions abbreviations beneath the annual plot. P40Figure14: The technical quality is poor.*

*P41Figure15: The technical quality is poor. Axis titles are insufficient (variable + unit missing).*

Figures 12-15 were replaced using higher quality plots and we included legends were applicable. For Fig 12 we now show the difference to ERA-Interim.

*P42Figure16 Does it make sense to give to exact date of the example - it is not comparable to reality/observed value. A hint that it is an artificial date would help.*

We added "...day (artificial date, not a real precipitation event), " to clarify this and removed the date from the caption.

*P43Figure17: 1900-2005*

For this figure, the whole time period 1900-2005 has been analyzed.

**Tables:** *Table1 is hard to read as the column for 'Description' is rather narrow, consider land- scape format if is possible and broaden the description-column.*

We changed the table to landscape and improved its formatting.

**Language/Spelling**

*P11L317: Please, consider rephrasing "is the single biggest contributor".*

We changed it to: "The energy sector is the largest contributor ..."

*P11L326: ".."*

Changed

*P13L400: "example" instead of "examples"*

Changed

*P15L451: Do not use small caps but write "Version 2.0"*

We changed it to "Apache License, Version 2.0"

**References in the answers:**

Déandreis, C., Braconnot, P., and Planton, S.: Impact du changement climatique sur la gestion des réseaux de chaleur. https://docplayer.fr/9496504-Impact-du-changement-climatique-sur-la-gestion-des-reseaux-de-chaleur.html, last access

24.02.2021, DALKIA, Étude réalisée pour l'entreprise DALKIA., 2014.

Lledo, L., Torralba, V., Soret, A., Ramon, J., and Doblas-Reyes, F. J.: Seasonal forecasts of wind power generation, Renew Energ, 143, 91-100, 10.1016/j.renene.2019.04.135, 2019.

Zhang, X. B., Alexander, L., Hegerl, G. C., Jones, P., Tank, A. K., Peterson, T. C., Trewin, B., and Zwiers, F. W.: Indices for monitoring changes in extremes based on daily temperature and precipitation data, Wires Clim Change, 2, 851-870, 10.1002/wcc.147, 2011.